

# Origin and Transformation of Ambient VOCs during a Dust-to-Haze Episode in Northwest China

Yonggang Xue[1,2,3,4], Yu Huang[1,2,3,4*], Steven Sai Hang Ho[1,2,5], Long Chen[1,2,3,4], Liqin Wang[1,2,3,4],
Shuncheng Lee[6], Junji Cao[1,2,3,4*],

[1]Key Lab of Aerosol Chemistry & Physics, Institute of Earth Environment, Chinese Academy of Sciences, Xi'an 7 0061, China
[2]State Key Lab of Loess and Quaternary Geology (SKLLQG), Institute of Earth Environment, Chinese Academy of Sciences, Xi'an 710061, China
[3]Shaanxi Key Laboratory of Atmospheric and Haze-fog Pollution Prevention, Institute of Earth Environment, Chinese Academy of Sciences, Xi'an 710061, China
[4]CAS Center for Excellence in Quaternary Science and Global Change, Xi'an, 710061, China.
[5]Division of Atmospheric Sciences, Desert Research Institute, Reno, Nevada, USA
[6]Department of Civil and Environmental Engineering, The Hong Kong Polytechnic University, Hung Hom, Hong Kong,
China

*Correspondence to*: Yu Huang (huangyu@ieecas.cn)

Junji Cao ( cao@loess.llqg.ac.cn)

**Abstract.** High contribution of secondary organic aerosol to the loading of fine particle pollution in China highlights the roles of volatile organic compounds oxidation. Therein, particulate active metallic oxides in dust, like $TiO_2$ and Fe ions, were

proposed to influence the photochemical reactions of ambient VOCs. A case study was conducted at an urban site in Xi'an, northwestern China, to investigate the origin and transformation of VOCs during a windblown dust-to-haze pollution episode, and the assumption that dust would enhance the oxidation of VOCs was verified. Local vehicle exhaust (24.76%) and biomass burning (18.37%) were found to be the two largest contributors to ambient VOCs. In the dust pollution period, sharp decrease of VOCs loading and aging of their components were observed. Simultaneously, the secondary oxygenated VOCs

fraction (i.e., methylglyoxal) increased. Source strength, physical dispersion, and regional transport were eliminated from the major factor for the variation of ambient VOCs. In another aspect, about 2 and 3 times increase of the loading of Iron (Fe) and titanium (Ti) was found in the airborne particle, together with fast decrease of trans-/cis-2-butene ratios which demonstrated that dust can accelerate the oxidation of ambient VOCs and formation of SOA precursors.




# 1 Introduction

Secondary aerosols are important components of fine particles in China, which could contribute to about 30 to 77 percent of $PM_{2.5}$ loading, therein, secondary organic aerosols (SOA) take about half of the loading (Huang et al., 2014). Guo et al. (2014) believed that gaseous emissions of volatile organic compounds (VOCs) and nitrogen oxides (NOx) were responsible for the large secondary PM formation. Laboratory experiments found OH-initiated oxidation of m-xylene could cause the coating thickness of black carbon, which further induced increase of particle size (1.5 to 10.4 times) and effective density (from 0.43 to 1.45 g $cm^{-3}$)(Guo et al., 2016).

Solid-gas heterogeneous reactions would cause the transformation of gaseous pollutants and change the property of particles (Zhang et al., 2000;Zhang et al., 2003;He et al., 2014). Recently, the oxidation of organic and inorganic gas on particles surface through the transitional-metal-catalyzed chain reaction was frequently found to play important roles on the transformation of ambient gas pollutants (Chu et al., 2019). Mineral dust is the most important sources of the transitional-metal, like Iron (Fe) and titanium (Ti), in the natural environment (Chen et al., 2012). In addition, mineral dust is one kind of the most abundant components of the global airborne PM, and about 1600 to 2000 Tg of mineral dust is transformed to aerosols annually from major deserts (Ginoux et al., 2001). Furthermore, the surface of mineral dust provides plenty of reactive sites for multiple atmospheric trace gas reactions (Cwiertny et al., 2008). As a result, dust was viewed to serve as catalyst for reactive gas, and modify the photochemical processes (Dentener et al., 1996;Dickerson et al., 1997).

With the controlled experiment of sulfate formation on mineral dust, Zhang et al. (2019) found that under appropriate humidity and particle acidity, surface transitional-metal-catalyzed chain reaction together with nitrate would highly accelerate the sulfate's formation on the surface of mineral dust (Zhang et al., 2019). In another aspect, gas-solid heterogeneous photochemical reactions of organic compounds were also reported on the illuminated surface of semiconductor metal oxides in the natural environment, in particular $TiO_2$ (Chen et al., 2012). Co-existence heterogeneous photochemical reactions of $SO_2$, $NO_2$ and VOCs on the surface of mineral dust were investigated in recent years. Both synergistic and suppress effects of VOCs on the formation of sulfate were found, which indicated the competition of reactive oxygen species and active sites between VOCs and inorganic gas pollutants (Chu et al., 2019; Song et al., 2019). In addition, oxidized products, like formate and acetate species, were observed in the co-existence reaction, which highlight the possibility of further oxidation of VOCs on the mineral dust (He et al., 2014). In northwestern China, dust from both local sources and long-range transport is one of the most important components of particulate matter of < 2.5 μm in diameter ($PM_{2.5}$) (Huang et al., 2014). Xi'an has a population of ~8 million (Feng et al., 2016). The sharp increase of vehicles and other human activities has led to high emissions of VOCs and $NO_x$ (Li et al., 2017). Observations showing simultaneous high dust loading and elevated VOCs and $NO_x$ concentrations suggest possible impacts from heterogeneous reaction on dust particles (Huang et al., 2014;Li et al., 2017). The present study was conducted to investigate the origin and transformation of ambient VOCs with severe dust-to-haze episode in winter. The transformation and the related chemical processing of



ambient VOCs and the related changes in the composition of PM$_{2.5}$ were studied, within typical windblown dust-to-haze episodes. The potential pathway of VOCs oxidation in the windblown dust-to-haze formation process was explored.

65 ## 2 Materials and Methods

### 2.1 Sampling site

An observation site (E 109°00′7", N 34°13′22") managed by Xi'an Jiaotong University was used in this study (Figure 1). All sampling equipment was deployed on the rooftop of a 15-m tall academic building. No obvious stationary pollution 70 sources were found nearby, and the location can be considered as a typical urban location in Xi'an (Zhang et al., 2015a).

### 2.2 Field Sampling

Severe dust-to-haze episode was observed in Xi'an and the surrounding areas from 8 November to 12 November in 2016, and sampling was continuously conducted during this period to investigate the chemical compositions of both VOCs and fine PM. A total of 57 non-methane VOCs species (i.e., C$_2$-C$_{12}$ saturated and unsaturated aliphatic and aromatic VOCs) 75 were sampled hourly into offline multi-bed adsorbent tubes; the measured 57 VOCs were defined as VOC$_{PAMS}$. The loaded tubes were analyzed using a thermal desorption and gas chromatography/mass spectrometry (TD-GC/MS) method. In previous developmental work, humidity and temperature during sampling were found to impact significantly on the analyses; for this study, all sample collections were made under optimized conditions(Ho et al., 2017;Ho et al., 2018). Sixteen airborne carbonyls (including mono- and dicarbonyls) were collected over diurnal cycles (i.e., 20:00–08:00 local time [LT] and 80 08:00-20:00 LT) by 2,4-dintrophenyhydrozine (DNPH) coated-cartridges. Detailed sampling and analytical procedures for VOCs and carbonyls can be found in previous publication (Ho et al., 2017;Dai et al., 2012).

PM$_{2.5}$ filter samples were sampled with mini-volume samplers (Model Mini-Vol, Air Metrics Co., Oregon, USA) with a flow rate of 5 L min$^{-1}$ (Cao et al., 2005). Fine PM was sampled by 47-mm quartz microfiber filters (Whatman QM/A, Maidstone, UK), and the filters were pre-heated at 900°C for 3-h before sampling. The loaded filters were transfered into 85 clean polystyrene petri dishes and stored in a freezer.

### 2.3 Chemical Analyses

Analytical procedures for VOC analysis have been described previously (Ho et al., 2017). In brief, the analytes in the adsorbent tubes were firstly desorbed in a thermal desorption unit (Series 2 UNITY-xr system with ULTRA-xr, Markes International, Ltd., UK) coupled to a GC/MS (7890A/5977B, Agilent Technologies, Santa Clara, CA, USA). The loaded 90 tube was transfered into the TD unit and blown with ultra-high purity He gas. The targeted VOCs were desorbed at 330°C


within 8 min,  and then refocused onto a cryogenic-trap (U-T1703P-2S, Markes) at −15°C.  The targeted VOCs were transfered to a cold GC capillary column head (Rtx®-1, 105 m × 0.25 mm × 1 mm film thickness, Restek Corporation, USA) at −45°C . The chromatographic condition could be found in our previous work (Ho et al., 2017).

For carbonyl compounds, the DNPH cartridges were firstly eluted with acetonitrile (HPLC/GCMS grade, J & K Scientific
Ltd., Ontario, Canada)(Dai et al., 2012). The extracts were analyzed with a typical high-pressure liquid chromatography (HPLC) system (Series 1200; Agilent Technologies) equipped with photodiode array detector. The column was matched with a 4.6 × 250 mm Spheri-5 ODS 5 μm C-18 reversed-phase column (Perkin-Elmer Corp., Norwalk, CT) (Dai et al., 2012;Ho et al., 2011).

The particulate organic carbon (OC) and elementary carbon (EC) were analyzed with a DRI model 2001 carbon analyzer
(Atmoslytic, Inc., Calabasas, CA, USA)(Chow et al., 2007;Chow et al., 1993). Anions ($Cl^-$, $NO_3^-$, and $SO_4^{2-}$) and cations ($Na^+$, $NH_4^+$, $K^+$, $Mg^{2+}$, and $Ca^{2+}$) in particles were determined in aqueous extracts of the sample filters. Detailed extraction and analytical procedures were presented in a previous publication (Zhang et al., 2011). The abundances of 25 particulate elements (Na, Mg, Al, Si, S, Cl, K, Ca, Sc, Ti, V, Cr, Mn, Fe, Co, Ni, Cu, As, Se, Br, Sr, Ba, Pb, Ga, Zn)  were measured by energy dispersive x-ray fluorescence (ED-XRF) spectrometry (Epsilon 4 ED-XRF, PAN alytical B.V., the Netherlands). The
X-ray source was matched with a  metal-ceramic X-ray tube with a Rh and Ag anode, and X-ray source was operated at a maximum current of 3mA, and the maximum accelerating voltage of 50kV (maximum power 15W).

### 2.4 Quality Control

The Minimum detection limits (MDLs) of the VOCs were in the range of 0.003–0.042 ppbv with a 3 L sampling volume (Table S1). The measurement precision  at 2 ppbv was ≤ 5%(Ho et al., 2017;Ho et al., 2018). Three field blank samples were
collected within each sampling day, and they were analyzed using the same procedures as those for the ambient air samples. Most target compounds were not detected in the field blanks, and propylene, benzene, and toluene were below their MDLs (< 0.23 g per tube and < 10% of the arithmetic mean of ambient samples). No breakthrough (~0%) was observed for $VOCs_{PAMS}$ except for $C_2$–$C_3$ hydrocarbons, which were < 10% when the air temperature was > 30°C. The MDLs for the carbonyl target compounds were between 0.009 to 0.067 ppbv at a sampling volume of 3.6 $m^3$. Negligible breakthrough (<
5%) was found under the sampling conditions and flow rates in the field.

## 3. Results and Discussion

### 3.1 Origins of ambient VOCs during Dust and Fine-particle Pollution Events

Similar levels of alkenes were seen at the cities of Beijing, Shanghai and Guangzhou comparing to that in the present study (Table S2, Ho et al., 2004;Liu et al., 2008b). Unexpectedly, the aromatics were slightly higher in Xi'an than that in Beijing,





and 50% higher than that in Guangzhou(Shao et al., 2009;Zou et al., 2015). Therein, ethylene, ethane, toluene, iso-pentane, propane, n-butane, iso-butane, propylene, n-pentane, and benzene were the most 10 abundant VOCs$_{PAMS}$. The high fractions of these markers reflect strong emissions from traffic and coal combustion or from biomass burning (Liu et al., 2008a;Ho et al., 2009;Huang et al., 2015;Fan et al., 2014;Zhang et al., 2015c). Previous studies found higher contributions of non-fossil sources to carbonaceous aerosols in Xi'an, as compared with Beijing (Ni et al., 2018). Generally, non-fossil emissions

mainly originate from biomass burning (Ni et al., 2018), and the higher contribution of non-fossil sources to carbonaceous in Xi'an would indicate remarkable biomass burning activities exist in Xi'an and the surrounding areas(Huang et al., 2014;Xu et al., 2016).

   Receptor models and correlations between individual VOCs have been used for source assessments. In this study, significant correlation ($R^2$=0.62, p<0.05, slope of 1.59) was found for a least-squares regression between toluene and

benzene (Figure S1). The ratio of toluene to benzene (T/B) ratio has been shown to different among combustion sources; for example, Liu et al. (2006) reported T/B ratios of 1.5-2.0 in gasoline-related emissions collected in a tunnel. In contrast, T/B ratios ranged from 0.23-0.68 and 0.13-0.71 for biomass burning and coal combustion, respectively (Zhang et al., 2015c). The T/B ratios in our samples ($R^2$=0.62, p<0.05, slope of 1.59) implied a strong impact from traffic on the ambient VOCs in Xi'an. Significant correlations (p<0.05) were observed among $C_3$-$C_5$ alkanes, with propane versus n-butane ($R^2$=0.75,

slope=0.91), n-pentane versus iso-pentane ($R^2$=0.85, slope=0.35), and trans-2-butene versus cis-2-butene ($R^2$=0.99, slope=0.84)(Figure S1). The observed ratio of propane to n-butane in Xi'an was 1.1, which is close to that (1.36) observed in the tunnel study cited above (Liu et al., 2008). High loadings of n-pentane and iso-pentane are indicative of unburned vehicular emissions, and Liu et al., (2008) reported a ratio of iso-pentane/n-pentane of 3 in tunnel air, which is consistent with the slope of 2.85 found in the present study. The results demonstrate strong impacts of from gasoline engines' exhaust

on the atmosphere of Xi'an.

   PMF model was used to identify the major pollution sources: the data input to the model were the mixing ratios and uncertainties in the VOCs mixing ratios for all valid samples collected during the study. Five sources were identified (Figure S2), and the detail process of source apportionment were given in the supporting information. Biomass burning and gasoline exhaust were the two most significant pollution sources, contributing 24.76% and 18.37%, respectively. The combustion of

LPG and CNG (24.67%), diesel exhaust (15.28%), coal combustion (16.92%) also were found to be important sources of ambient VOCs (Figure S2). Biomass is commonly used for heating and cooking in rural areas of the basin in winter due to its low cost compared with natural gas and electricity. Consistent with our results, previous studies found high contribution of biomass burning and gasoline exhaust to the organic aerosol in Guanzhong Basin(Cao et al., 2005).

   Clear air conditions occurred at the beginning of the sampling period, but severe dust and fine-particle pollution events

were observed afterward. The high dust event was defined by loading of particulate matter ≤10 μm in diameter (PM$_{10}$) between 300 and 500 μg m$^{-3}$, and these conditions occurred from 12:00 LT on 9 November to 13:00 LT on 10 November. The abatement of dust before the fine particle pollution event is referred to as the transition period (i.e., PM$_{10}$ < 300 μg m$^{-3}$


and PM$_{2.5}$ < 100 μg m$^{-3}$). The loading of PM$_{2.5}$ subsequently increased, and heavy fine particle pollution (PM$_{2.5}$ > 100 μg m$_{-3}$) occurred after 18:00 LT on 11 November.

Ratios of individual VOCs can be used to identify the origins of the compounds and to study atmospheric aging processes due to the special composition of VOCs in a typical source and the different lifetime of VOCs species (Xue et al., 2017; Zhang et al., 2015c). In addition, influences from meteorological variation and atmospheric transport also need to be considered when characterize the potential sources of the compounds in ambient air. To investigate the impacts of air mass transport on VOCs concentrations, we calculated air-mass back trajectories using the NOAA HYSPLIT model for the dust

event (Figure S4a) and for the fine-particle pollution episode (Figure S4b). The trajectories were calculated at an arrival height of 500 m above ground at the observation site. In view of the short atmospheric lifetimes of VOCs (for example, isoprene, ~1.4 h; propylene, ~5.3 h; toluene, 2.1 d) (Atkinson and Arey, 2003), 24-h back trajectories were used for this assessment.

    Clear different air masses back trajectories and VOCs ratios were observed between dust pollution and haze pollution

periods. From  9 November to 10 November (in dust pollution period), the air mass reaching Xi'an passed over areas to the west of the city (i.e., Gansu Province and Ningxia Autonomous Region) through long range transport; after 11 November (formation of haze), the transport of air mass was mainly limited to areas around southern Xi'an. Differences in the chemical compositions of ambient VOCs in the dusty versus in the haze event can clearly be seen (Figure 2) in the ratios of toluene to benzene (T/B) and m, p-xylene to ethylbenzene (X/E). During the clear and dusty periods, the T/B and X/E ratios varied

significantly with time of day; that is, the highest values for T/B (4.5–9.0) and X/E (0.98–1.05) were seen during rush hour (07:00–09:00 LT and 17:00–19:00 LT), while the lowest values (0.50–1.95 for T/B, and 0.89–0.96 for X/E) occurred in the early afternoon (i.e., 14:00–15:00 LT). The timings of the high T/B and X/E ratios suggest that fresh emissions from local traffic were the major source for the ambient VOCs, and this implies that long-range transport did not have a strong impact on the ambient VOCs during the clear or dusty parts of the study (Ho et al., 2004;Liu et al., 2008a). While during the

transitional and fine PM pollution period, both T/B and X/E varied but at relatively lower values compared with the earlier parts of the study (T/B, 3.33±1.97, 2.21±0.86, 1.91±0.74, 2.01±0.56 in clear, dust, transitional and fine particle pollution periods, respectively; X/E, 1.00±0.05, 1.05±0.12, 0.93±0.17, 0.95±0.13 in clear, dust, transitional and fine particle pollution periods, respectively). These synchronous lower values of T/B and X/B in transitional and fine particle pollution periods were indicative of aged air masses (Zhang et al., 2015c;Xue et al., 2017;Warneke et al., 2013).

Variations in the air mass transport pathway, and T/B or X/E during different sampling periods (clear, dust, transitional, fine particle pollution) confirmed that ambient VOCs were fresh in the clear and dust periods, but relatively aged during the transitional and fine particle pollution periods(Zhang et al., 2015c;Xue et al., 2017;Warneke et al., 2013). This indicates that the long-range transport of air mass had a relatively weak influence on the ambient VOCs even during the high dust period. Otherwise, composition of ambient VOCs should be relative aged due to long exposure time with dust transport. Indeed,

emissions from local vehicular exhausts and biomass burning in Xi'an and the surrounding areas were the main contributors to ambient VOCs throughout our study.



### 3.2 Transformation of VOCs between Dust and Fine Particle Events

With the shading of dust, level of ambient VOCs decreased with time, and the low concentrations (8.3 to 33.9 ppbv) were observed from 13:00 LT on 10 November and 01:00 LT on 11 November (Figure 3). During the fine particle pollution period (12–13 November), the $\Sigma$ VOCs$_{PAMS}$ increased, reaching an average of 38.0 ppbv in the last 24 h, compared with 19.0 ppbv in the transitional period and 21.5 ppbv in the first 12 h of the fine particle pollution episode (Figure 3). This buildup of VOCs can be explained by weak dispersion and relatively shallow boundary layers (400–1000 m) during the event (Figure 3). In addition, during this transition period, much lower ratios of T/B and X/E were observed in comparison with those in other periods (as mentioned in the part 3.1.2). We propose the possibility that windblown dust which include sustainable TiO$_2$ can influence the atmospheric photochemistry of VOCs, which would accelerate the oxidation of ambient VOCs (Chu et al. 2019;Nie et al., 2014).

While changes in the emission sources and their strengths, physical dispersion, regional transport, and aging of air masses all could affect VOC levels and composition (Xue et al., 2013;Xue et al., 2017). As a result, to evaluated aging of ambient VOCs in different period, the impact of dust on the transform of ambient VOCs, and the relative processes, the mentioned factors should be fully considered.

Significant variations in ambient VOCs were observed between the high dust and PM events. That is, during the clear and dusty periods—and similar to the trends in T/B and X/E ratios—peaks in $\Sigma$ VOCs$_{PAMS}$ were seen from 17:00 to 20:00 LT and from 09:00 to 12:000 LT (Figure 3), again highlighted the impacts of local traffic emission (Liu et al., 2008a;Huang et al., 2015). Typically, 1,3-Butadiene is often used as marker of gasoline-powered motor vehicles (Huang et al., 2015); in contrast, ethane is key chemical marker for biomass and coal combustion (Liu et al., 2008a). Time series plots of 1,3-butadiene and ethane (Figure S5) show that peaks in 1,3-butadiene mostly occurred during rush hour, while higher concentrations of ethane were seen during the night. These results support the conclusions that there were strong impacts from gasoline-powered motor vehicles in the daytime and from biomass burning or coal combustion for heating at night.

While with the shading of dust transport, shallow boundary layers were observed in the transitional period. For the clear and dust transport period, the boundary layer between 08:00 to 14:00 was relatively deep (1150–1500 m). In contrast, the boundary layer height decreased sharply to < 800 m on 11 November in transitional period. This limited the possibility that diffusion caused the sharp decrease of ambient VOCs in the transitional period. In addition, winter heating activities was relatively active because of low temperatures during the transitional period, and this limited the possibility of reduced emission amounts. Hence the variations of sources strength and physical dispersion are eliminated from the major factor caused the extremely low concentration and relative aged composition of ambient VOCs.

Significant impacts of air mass input was eliminated. Input of air mass would certainly cause the variations of VOCs' composition and loading(Xue et al., 2014). In the present study, long range transport of air masses had limited impacts on the characteristic of ambient VOCs during the sampling period. In another aspect, relative active VOCs would be firstly degraded, hence composition of ambient VOCs would be aging with long range transport (Ho et al., 2009;Xue et al., 2017).



While in the present study, as mentioned above, composition of ambient VOCs was relative fresh under long range transport of air mass (within dust transport). In contrast, VOCs composition was relatively aged under the air mass that limited with Xi'an and the surrounding area (transitional period). This phenomenon indicated that regional transport cannot be the major factor inducing the relative aged composition and excess low loading of the ambient VOCs in the transitional period.

      Synchronous changes of the VOCs isomeride were found in the windblown dust-to-haze episode, which supplied the
evidence of the accelerated photochemistry reactions. In the present study, we found fast decrease of trans-/cis-2-butene ratio within dust transporting, which confirmed the accelerated photochemical reactions of ambient VOCs (Figure 4). Trans-2-butene and cis-2-butene are two isomerides that mostly emitted from same sources (Zheng et al., 2017;Zhang et al., 2015b). While trans-2-butene has higher photochemical reactions rate with OH radical in the atmosphere ($k_{OH}$ $6.40\times10^{-11}$ $s^{-1}$) than cis-2-butene ( $k_{OH}$ $5.64\times10^{-11}$ $s^{-1}$ ) (Perring et al., 2013), hence trans-/cis-2-butene ratio would decrease with the
photochemical reactions (Zhang et al., 2015b). Firstly, relative higher trans-/cis-2-butene ratios were observed in the rush hours (evening rush hours 17:00-20:00, morning rush hours 07:00-10:00)(Figure 4), which indicated fresh emission from local traffic activities (Zhang et al., 2015b). In addition, sharp decrease of trans-/cis-2-butene ratio was observed from late half of windblown dust period to the end of transitional period (Figure 4). The quickly shrinking of trans-2-butene comparing to cis-2-butene in the dust pollution period indicated that oxidation of ambient VOCs was accelerated in the
period with high loading of the suspending dust particles(Zhang et al., 2015b).

      Significant increase of particulate active metals was found in dust pollution period, which further verified the promotion of dust on the heterogeneous reactions. Previous study found that mineral dust can affect the chemistry of the atmosphere by scavenging gaseous compounds (Zhang et al., 2000;Chen et al., 2012); it can also promote heterogeneous reactions of atmospheric substances, including VOCs, because the particle surfaces can provide sites for photo-catalytic reactions
(Cwiertny et al., 2008;Ndour et al., 2009). In the present study, ferrum (Fe) and titanium (Ti) contents of the particulate increased significantly within the period with dust transport through (Figure 5). In detail, content of Fe increased from 19.3 $\mu g$ $m^{-3}$ in clear days to 40.8 $\mu g$ $m^{-3}$ in dust pollution days, and the content of Ti increased from 0.92 to 2.98 $\mu g$ $m^{-3}$. Hence, huge increase of the Ti and Fe concentrations in particulate phase during the period of dust pollution days could possibly promote the gas-solid photochemical reaction of the ambient VOCs, which would reasonably ascribe the relative low level
and aged composition of ambient VOCs in this period(Chu et al., 2019; He et al., 2014;Song et al., 2019).

### 3.3 Variation of carbonyl compounds between dust to fine particle pollution periods: further formation oxygenated VOCs with aging of primary VOCs

      Aging of primary VOCs and formation of carbonyl compounds were observed synchronously, as the fine-particle pollution event developed (Figure 3; Figure 6a). As discussed above, relatively low T/B and X/E values were observed
during the transitional and fine PM periods after the dust event (Part 3.2). In our study, the carbonyl levels increased after the clear and dusty periods, and the highest levels were seen during the fine particle pollution event (Figure 6a). Carbonyl compounds are produced from both the primary sources and form through secondary processes (Dai et al., 2012;Duan et al.,





2012), and we found higher carbonyl concentrations during daytime than at night (Figure 6a). This is consistent with previous studies in Xi'an (Dai et al., 2012), which confirmed the secondary formation of carbonyl compound under sunlight illumination.

Methylglyoxal is generally considered to be a secondary species, while acetone is mainly from primary emissions; the ratio of acetone to methylglyoxal (A/M) has been used as an indicator of air mass aging(Dai et al., 2012;Liu et al., 2006). In the present study, A/M ranged from 12 to 14 during the clear and first half of dusty periods but then dropped sharply and stayed between 6 and 9 during the later parts of dust pollution period, transitional and the high PM event (Figure 6a). Increases in the abundances of carbonyl compounds and lower A/M ratios suggested relatively stronger aging of the air masses, this is further evidence of fast degradation of VOCs in the late half of the windblown dust event, and the primary VOCs were oxidized and served as precursors of SOA. In consequence, composition of particles changed with oxidation of ambient VOCs across the sampling periods.

### 3.4 Variations of PM$_{2.5}$ Chemical Composition during Dusty and Fine PM Pollution Periods

Significant variations of water-soluble inorganic ions, OC, and EC were observed diurnally and between dust and fine particle pollution events (Figure 6b, c). For instance, the concentrations of NO$_3^-$ were relatively high in the daytime, while K$^+$ was more abundant at night. The diurnal cycles can be explained by the formation of secondary particles through photochemical processes during the daytime and by the impacts from biomass and coal burning for heating at night(Dai et al., 2012;Zhang et al., 2018;Cong et al., 2015). The concentrations of Ca$^{2+}$, Mg$^{2+}$, and Na$^+$, which are typically associated with dust in inland areas (Wu et al., 2011), increased sharply during the dusty period, and then declined rapidly afterwards.

As discussed, the apparent contribution of VOCs to the formation of SOAs increased when the dusty conditions transitioned into a fine-particle pollution event. Temporal changes in the chemical composition of PM$_{2.5}$ are consistent with this suggestion. During the fine-particle pollution period, both the concentrations of secondary ions, particularly NO3-, increased as the haze event developed. A similar trend was seen for OC (Figure 6b), and this we attribute to the formation of SOAs. Combined with the findings regarding the compositions of VOCs and PM$_{2.5}$, these results indicate that the reactions of VOCs led to the formation of SOA, and in so doing contributed to the fine particle pollution.

### 4. Conclusion

SOA is key component in the loading of PM$_{2.5}$ across China, and the oxidations of ambient VOCs were proved to play key roles on the formation of SOA. Natural particulate metallic oxides, like TiO$_2$, were found to promote the gas-solid heterogeneous reactions of ambient VOCs. Comprehensive field work was carried out to investigate the origin and transform of VOCs within the dust-fine particles pollution periods in winter with the city of Xi'an. And the assumption of promotions of dust on the heterogeneous reactions of VOCs was further verified. Local vehicle exhaust and heating activities were found to be the most important sources of the ambient VOCs in Xi'an within winter, while long range transport air mass has



limited impacts. Within the period of dust transport, loading of ambient VOCs decreased sharply from the late half period,
and the lowest concentration was observed in the transitional period, in accordance with aging of primary VOCs. In addition,
loading and proportion of secondary VOCs in gaseous phase and secondary ions and organic carbon in particulate phase
increased with the aging of primary VOCs. Source strength, physical dispersion, and regional transport were eliminated from
the major factor for the variation of the ambient VOCs. On another aspect, sharp increase of active metals concentrations (Ti
and Fe) and fast decrease of trans-/cis-2-butene ratio were observed from the late half of dust transport period. In
consequence, we conclude that windblown dust might accelerate the gas-solid heterogeneous reactions of atmospheric VOCs,
and further induced the formation of SOA precursors.

*Data availability.* All of the research data have been included in the supplement.

*Supplement.* The following information is provided in the Supplement: Sampling procedures, Chemical Analysis, Source
characterization, Figure S1-S5, Table S1-S2.

*Author contributions.* YX designed the study. YX and YH wrote the paper. SH, JC and SL revised the manuscript, LC and
LW analyzed the data. All authors reviewed and commented on the paper.

*Competing interests.* The authors declare that they have no conflict of interest.

*Acknowledgements.* This research was financially supported by the National Key Research and Development Program of
China (Grant No. 2016YFA0203000), and the National Science Foundation of China (Grant No. 41701565, 21661132005,
41573138). Yu Huang was also supported by the "Hundred Talent Program" of the Chinese Academy of Sciences. The data
used are listed in the supplements.

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




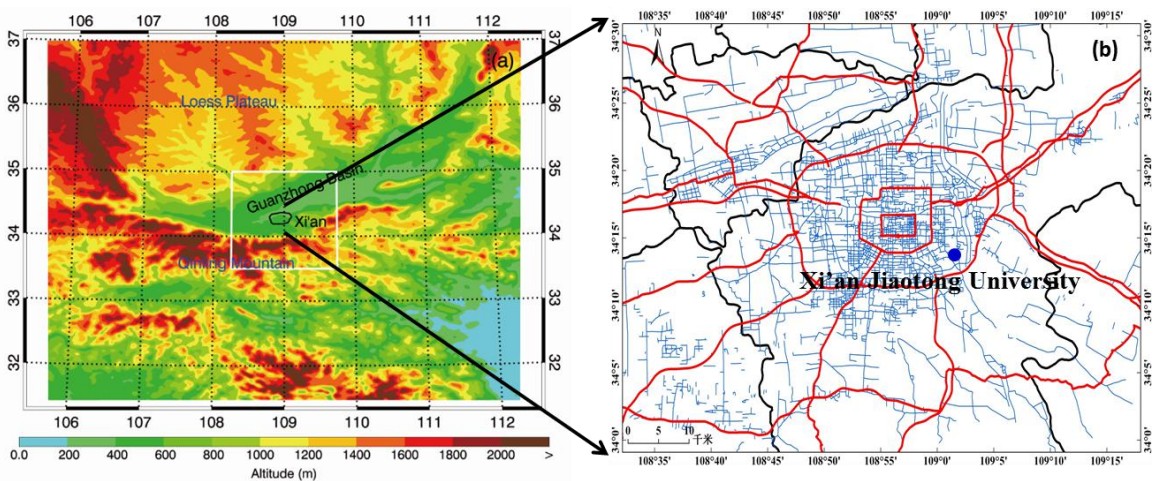

**Figure 1: Regional and local maps of the study area, (a) Regional map showing the location of Xi'an and the surrounding geography; (b) local map of Xi'an showing the sampling site (blue dot), main roads (red lines), and secondary roads (blue lines).**









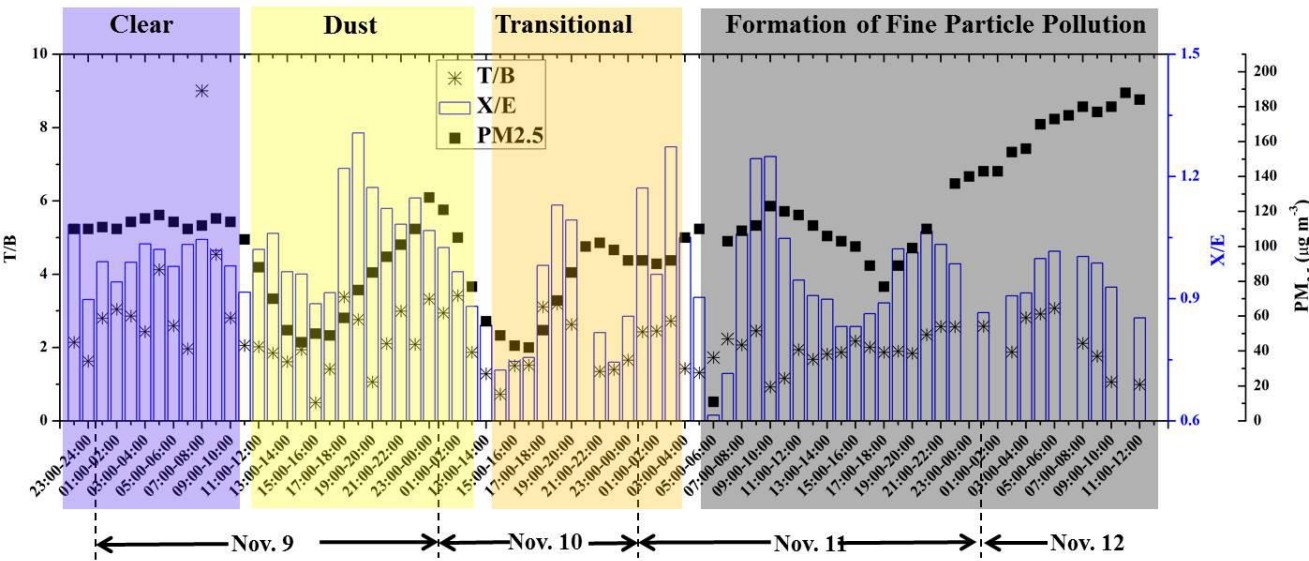

**Figure 2: Variations in the ratios of indicator volatile organic compound (VOC) species (toluene/benzene [T/B], and m-,p-xylene/ethylbenzene [X/E]) and fine particle loadings during the study period.**






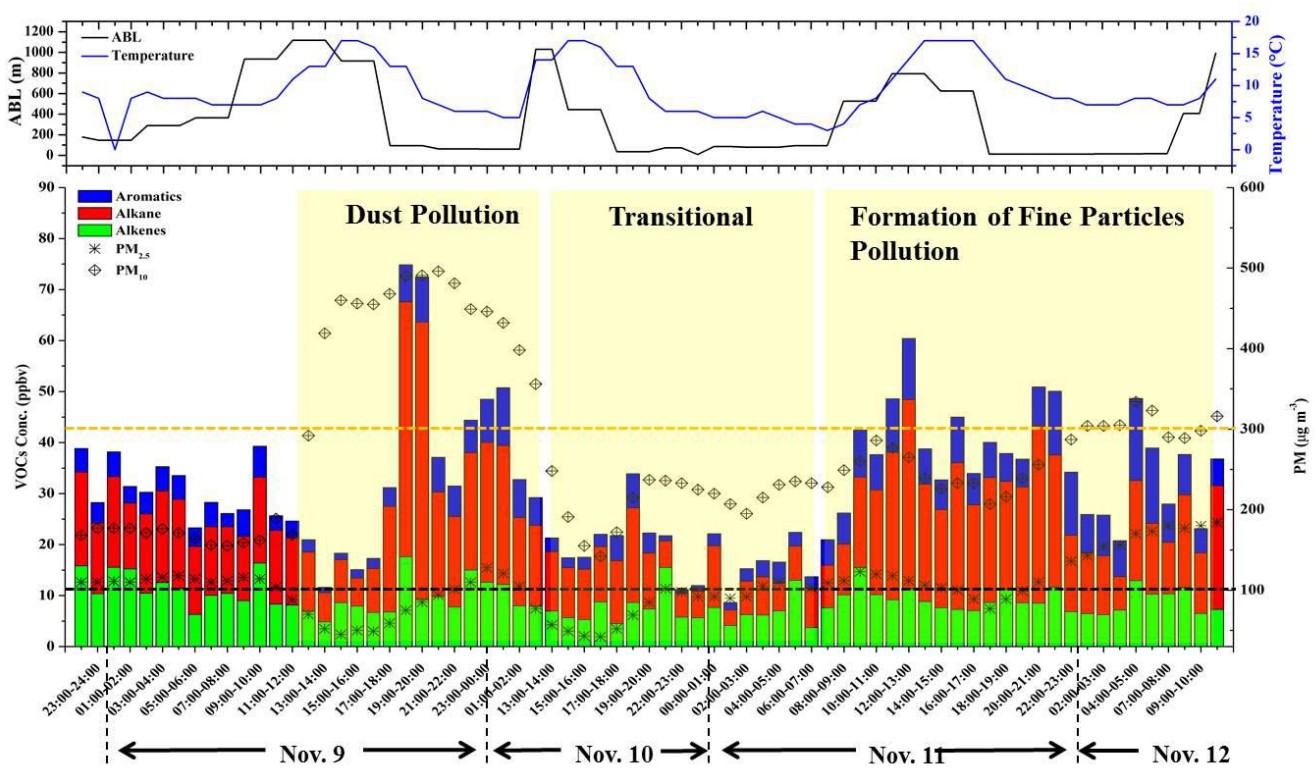

**Figure 3: Temporal variations in volatile organic compound (VOC) concentrations and particle levels during the sampling period (9–13 November 2016).**




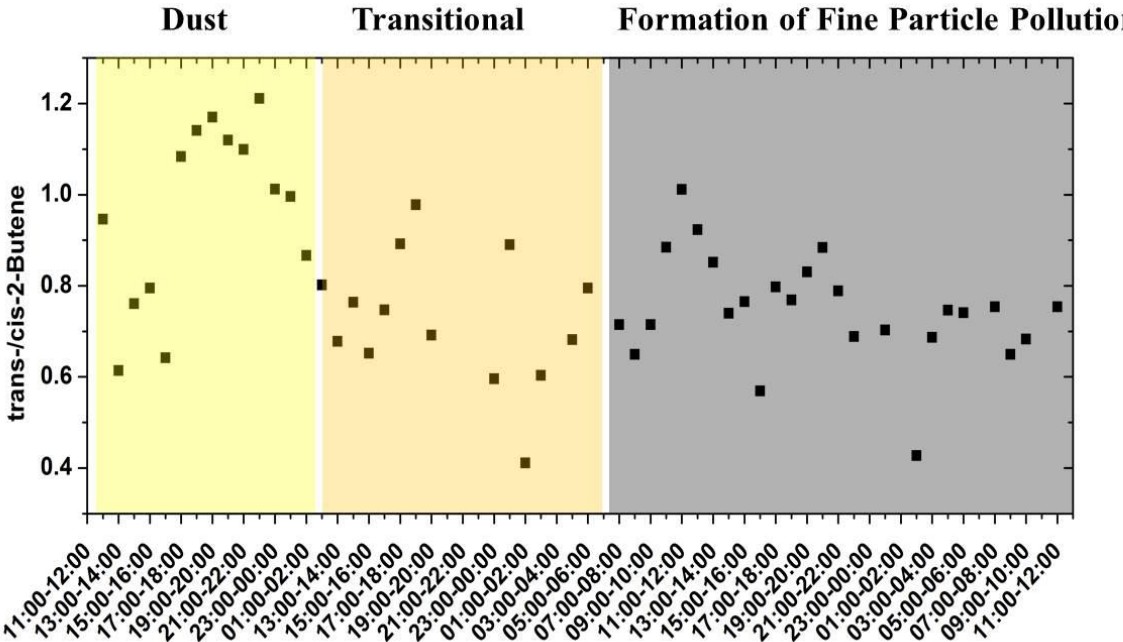

**Figure 4: Temporal variation of trans-/cis-2-butene ratio in the dust-transitional-fine particle pollution period.**








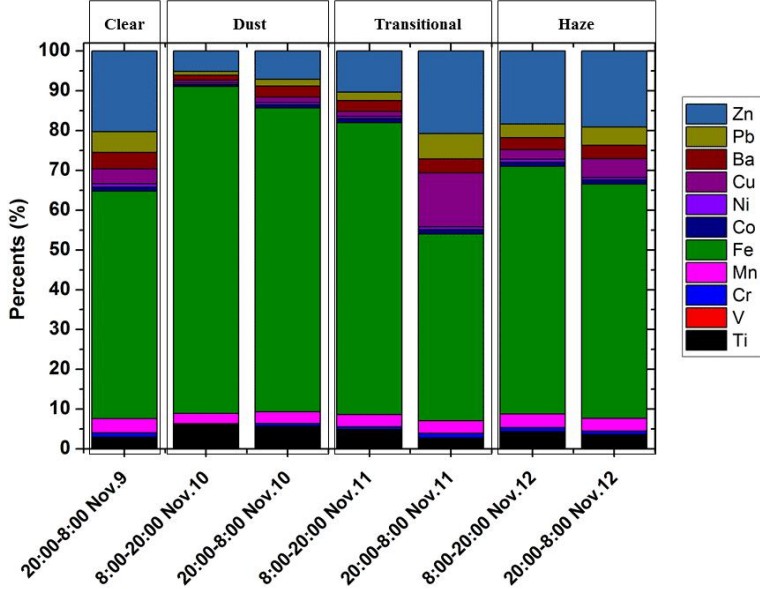

**Figure 5: Composition of selected metallic elements in the PM2.5 samples.**









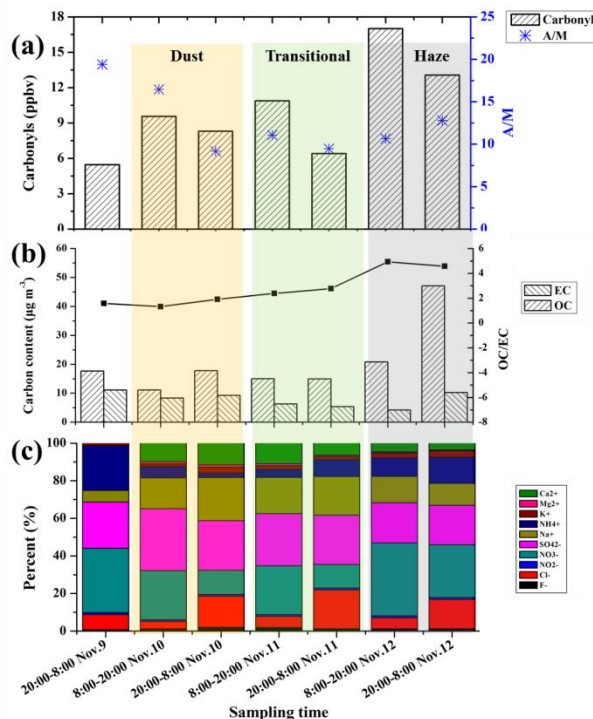

**Figure 6:** Variations in (a) the mixing ratios of 17 carbonyl compounds and acetone to methylglyoxal (A/M) ratios in the gas phase, (b) particulate carbon fractions, (c) and particulate water-soluble ions during the study period.