# Peer review of "Origin and Transformation of Ambient VOCs during a Dust-to-Haze Episode in Northwest China"

_Atmospheric Chemistry and Physics, 2019_

## Short Comment (SC1) · 23 Dec 2019

The current manuscript organized well. But some places may be improved in future:

1) Why did the authors choose the period of dust-to-haze episode to observe? How did it relate with the reaction of aerosols? Please describe the relationship between the scientific issue and the research design.

2) In the Results and Discussions, the basic results, such as the levels of pollutants and the ratios of compounds, were not shown the manuscript. It is strange for readers that the sections starts with the sentences like "the concentrations in Xi'an are similar

with Beijing" or "the T/B ratios in this study indicated that..."

3) I don't think it is reasonable to use HYSPLIT model in L157-186. It had discussed that the pollution come from local emission in the above sections.

4) The abstract and Introduction highlighted the effect of metallic oxides on the photo-chemical reactions of VOCs. But in the manuscript, this situation wasn't discussed in details, and even no relationship figures of VOCs and metals were shown.

---

## Referee Comment (RC1) · Anonymous Referee #1 · 6 Jan 2020

This manuscript presents a case study on the origin and transformation of ambient volatile organic compounds (VOCs) during an episode of dust-to-haze in a city in northwestern China. It presents the variations of VOCs, oxygenated VOCs, PM2.5 and its chemical components (i.e., ions, carbonaceous fractions and elements) during this interesting transition of dust-to-haze. It also highlights the possible reactions of VOCs under the influence of source contribution, ambient conditions as well as the heterogeneous reactions promoted by particle elements. The manuscript is well prepared and fits the scope of the journal. It could be considered for publication after the following concerns are clarified.

[Figure]

(1) The authors suggested the formation of secondary organic aerosols by the reactions of VOCs during the haze period. It could be true due to the significant increases of organic carbon during the haze period. However, I would suggest more efforts should be made to clarify this point. Somehow I would also suggest that it might be necessary to be highlighted in the text as well as in the title.

(2) In the part of Section 3.2, I found it was quite difficult to read through this part. The influences of sources (seems not explicit), boundary layer, long-range transport and photochemical reactions were discussed but not in a clear manner, which made me very difficult to follow the conclusions.

(3) L224-226: The variations of the ratios of trans/cis-2-butene are discussed here. The finding/suggestion by the authors seems not really true if we have a close look at Fig. 4. For example, the increase in the rush hour seems not always the case during this observation. The increases could also be observed during the time periods of 19:00-22:00 and 11:00. In this case, it should be more careful to draw any conclusions e.g. to highlight the importance of photochemical reactions and promotion of dust particles.

(4) As mentioned above, the linkage of VOCs to PM2.5 and its chemical components in Section 3.4 should be discussed in a more clear and explicit manner. I think catalysis of particle metal could be very important during the formation of haze and it should be studied here. By the way, I found Cl- was elevated at night (see Fig. 6) and it might show the contribution of biomass burning in northern areas of China.

(5) L102-106: Quartz filter seems not an ideal filter media for XRF analysis due to the uneven surface. The authors should present more details of their XRF analysis.

(6) minor mistakes: L158: It should be "the potential sources…are characterized". L241: "through"should be deleted.

---

## Author Comment (AC1) · 8 Jan 2020

We are highly grateful to the reviewer, and your comment is highly helpful to further improvement of our manuscript and further study.

Comment 1. Why did the authors choose the period of dust-to-haze episode to observe? How did it relate with the reaction of aerosols? Please describe the relationship between the scientific issue and the research design.

Response: Dust is one of the most important sources of airborne metallic oxides in the natural environment, and the previous studies found explosive growth of ultrafine

particles following dust episode, which implied positive effect of dust on the formation of fine particles. In addition, controlled experiments in chamber found surface transitional-metal-catalyzed chain reaction would highly impact formation of organic and inorganic aerosols on the surface of mineral dust (this was introduced in the part of introduction). Hence, dust was viewed to highly impact the transformation of organic and inorganic gas in the atmosphere. While few field work on the transformation of ambient VOCs in the dust event were found in the previous studies to support the assumption that dust would impact the transformation of VOCs in the ambient VOCs. The present study was designed to investigate the transformation of VOCs in dust event, and this work might helpful on further understanding of the impacts of dust on the transformation of VOCs and its further impact on the formation of fine particles.

Comment 2. In the Results and Discussions, the basic results, such as the levels of pollutants and the ratios of compounds, were not shown the manuscript. It is strange for readers that the sections starts with the sentences like "the concentrations in Xi'an are similar with Beijing" or "the T/B ratios in this study indicated that. . ."

Response: This comment is fully accepted. In our manuscript, Table S2 were given in the supporting information, and the level and composition of ambient VOCs was summarized in this table. We will further improve our manuscript in this part, and the detail description of VOCs level and composition will replenished in the next version of manuscript.

Comment 3. I don't think it is reasonable to use HYSPLIT model in L157-186. It had discussed that the pollution come from local emission in the above sections.

Response: Thanks very much for this comment. In the present study, origin of the ambient VOCs was analyzed in two aspects. In the first aspect, emission source types of VOCs were estimated with PMF model. In another aspect, effect of transport or local emission on the ambient VOCs were evaluated based on the variation of VOCs ratios and air mass transport. In this part, back trajectory of air mass in different period of

sampling were used. Hence, the author believe hYSPLIT model should be useful in this part. And we decide to firstly remove the description of back trajectory of air mass to supporting information, if this decision will not be accept, we will delete this part.

Comment 4. The abstract and Introduction highlighted the effect of metallic oxides on the photochemical reactions of VOCs. But in the manuscript, this situation wasn't discussed in details, and even no relationship figures of VOCs and metals were shown.

Response: The objective of this work is to supply a reference of evidence the effect of dust on the transformation ambient VOCs and their further impact on the formation of secondary aerosol. While we believe in this study, our field work could just supply a solid transformation of ambient VOCs in the dust episode, which should be just a lateral evidence of particulate metallic oxides' catalysis effect on the oxidation of VOCs, because too many factors impact the variation of level and composition of ambient VOCs. In addition, because the sampling time of VOCs and particles were not synchronous. The VOCs were sampled hourly, while the particles were only collected with two samples in 24 hours. As a result, we can not get a correlation analysis result of particulate metals verse VOCs. And in the following study, we will carry out fully chamber work on the transformation of VOCs on the dust surface that might supply direct evidence of dust catalysis effect on the transformation of VOCs.

---

## Author Comment (AC2) · 16 Jan 2020

We are highly grateful to the reviewer, and your comments are highly helpful to further improvement of our manuscript and further study. The comments were fully considered, and related improvement were finished in the new version of manuscript.

Comment: This manuscript presents a case study on the origin and transformation of ambient volatile organic compounds (VOCs) during an episode of dust-to-haze in a city in northwestern China. It presents the variations of VOCs, oxygenated VOCs, PM2.5 and its chemical components (i.e., ions, carbonaceous fractions and elements) during this interesting transition of dust-to-haze. It also highlights the possible reactions

of VOCs under the influence of source contribution, ambient conditions as well as the heterogeneous reactions promoted by particle elements. The manuscript is well prepared and fits the scope of the journal. It could be considered for publication after the following concerns are clarified.

(1) The authors suggested the formation of secondary organic aerosols by the reactions of VOCs during the haze period. It could be true due to the significant increases of organic carbon during the haze period. However, I would suggest more efforts should be made to clarify this point. Somehow I would also suggest that it might be necessary to be highlighted in the text as well as in the title.

Response: The suggestion is highly accepted, the point of formation of SOA by the VOCs during the haze period if replenished in the part of 3.4. Seen in Line 277-282(in the new version), "A similar trend was seen for OC (Figure 6b), and content of particulate OC increased from 11.1 since dust event period to 47.1 in the haze period. In another aspect, the ratio of OC/EC increased from 1.3 to 4.9 in the dust-to-haze episode. The previous studies on the characterization of particles from traffic emission reported OC/EC values in the range of 0.28 to 0.92 in the diesel vehicles, and the OC/EC values were reported >2 in the gasoline vehicles. In addition, the OC/EC was reported in the range of 0.9 to 1.6 in the urban region in the city of Guangzhou. In the present study, the consistent increase of OC/EC would prove the formation of SOA in the dust-to-haze episode. "

(2) In the part of Section 3.2, I found it was quite difficult to read through this part. The influences of sources (seems not explicit), boundary layer, long-range transport and photochemical reactions were discussed but not in a clear manner, which made me very difficult to follow the conclusions.

Response: the suggestion is accepted, and this part is reorganized, as seen in section 3.2. As seen in line 201-202, "To evaluate the impact of sources types on the variation of VOCs in the dust-to-haze episode, diurnal variation of VOCs was depicted. " was

replenished. Line 209-212, "In addition, winter heating activities was relatively active because of low temperatures during the transitional period, and this limited the possibility of reduced emission amounts. Hence the variations of sources strength was eliminated from the major factor caused the extremely low concentration and relative aged composition of ambient VOCs. " was replenished. Line 213, "Variation of physical dispersion was also eliminated." was replenished.

(3) L224-226: The variations of the ratios of trans/cis-2-butene are discussed here. The finding/suggestion by the authors seems not really true if we have a close look at Fig. 4. For example, the increase in the rush hour seems not always the case during this observation. The increases could also be observed during the time periods of 19:00-22:00 and 11:00. In this case, it should be more careful to draw any conclusions e.g. to highlight the importance of photochemical reactions and promotion of dust particles.

Response: In the present study, fast decrease of the ratios of trans/cis-2-butene in the dust-to-haze episode. In the new version of manuscript, correlation analysis of trans/cis-2-butene verse sampling time was done in the dust-to-haze episode, and significant correlation were observed with a R2 of 0.6, and with a slope of 0.027/h, in spite of the interference of rush hour emission. As mentioned in the comment, scatter peak of the ratios of trans/cis-2-butene were observed in the periods of 19:00-22:00 and 11:00. In our previous work of the characterization of VOCs in the roadside environment, peak of ambient VOCs were sometimes observed in 8h-12h, which indicated tardive rush hours in the city of Xi'an. In addition, VOCs peaks were always observed in the midnight, and relative high density of high duty trucks that were used for Construction waste collection. Hence, the traffic emission of VOCs in the city of Xi'an is relative specific, which would be the reason of scatter peaks of the ratios of trans/cis-2-butene were observed in the periods of 19:00-22:00 and 11:00.

(4) As mentioned above, the linkage of VOCs to PM2.5 and its chemical components in Section 3.4 should be discussed in a more clear and explicit manner. I think catalysis of particle metal could be very important during the formation of haze and it should be

studied here. By the way, I found Cl- was elevated at night (see Fig. 6) and it might show the contribution of biomass burning in northern areas of China.

Response: we also believe the catalysis of particle metal could be very important during the formation of haze, and the linkage if VOCs to PM2.5 was further replenished in the section 3.4 (line 277-282). As mentioned by the reviewer, the content of Cl- was indeed higher in the night than daytime, this should be ascribed to heating activities (biomass or coal combustion). and this description was replenished in line 269.

(5) L102-106: Quartz filter seems not an ideal filter media for XRF analysis due to the uneven surface. The authors should present more details of their XRF analysis.

Response: In the present study, Quartz filter were used for PM2.5 samples collection, due to the demand of analysis of carbon content and water soluble ions. While quartz filter can not be used to analyze the content of some element with XRF, like Si. As a result, the content of particulate Si in the present study was not reported.

(6) minor mistakes: L158: It should be "the potential sources. . .are characterized". L241: "through"should be deleted.

Response: the comment was fully accepted, and further improvement was done in line 158 in new version of manuscript.
* * *
[Figure]

**Fig. 1.**

---

## Referee Comment (RC2) · Anonymous Referee #2 · 18 Jan 2020

The manuscript entitled "Origin and Transformation of Ambient VOCs during a Dust-to-Haze Episode in Northwest China" discussed the characteristics of ambient VOCs in a northwestern city in China, and the transformation of VOCs during a dust-to-haze episode was explored in this study. Generally, the paper is well organized and presented, and shows the possibility of VOC transformation through heterogeneous reactions during the episode. The paper can be considered for publication after the following minor revisions are made. Some specific comments are listed below.

Line 73: Change "sampling" to "samples"

Line 82: Change "with" to "by"

[Figure]

Line 139ïijŽThe last sentence is not clear. The contribution of gasoline vehicular emissions on ambient VOCs should be pointed out.

Line 150: It should be "aerodynamic diameter"

Line 228: Is the difference of photochemical reactions rate with OH radical between trans-2-butene and cis-2-butene large enough to compare?

Line 278-281: The first two sentences are not necessary in conclusion part, moreover, key conclusions should be presented by supporting data.

---

## Author Comment (AC3) · 20 Jan 2020

comment: The manuscript entitled "Origin and Transformation of Ambient VOCs during a Dust- to-Haze Episode in Northwest China" discussed the characteristics of ambient VOCs in a northwestern city in China, and the transformation of VOCs during a dust-to-haze episode was explored in this study. Generally, the paper is well organized and pre- sented, and shows the possibility of VOC transformation through heterogeneous re- actions during the episode. The paper can be considered for publication after the following minor revisions are made. Some specific comments are listed below.

1. Line 73: Change "sampling" to "samples"

[Figure]

Response: this sentence was changed to "Severe dust-to-haze episode was observed in Xi'an and the surrounding areas from 8 November to 12 November in 2016, and samples was continuously collected during this period to investigate the chemical compositions of both VOCs and fine PM." Line 81 in new version.

2. Line 82: Change "with" to "by"

Response: this sentence was changed to "PM2.5 filter samples were sampled with mini-volume samplers (Model Mini-Vol, Air Metrics Co., Oregon, USA) by a flow rate of 5 L min-1", line 93 in new version.

3. Line 139 The last sentence is not clear. The contribution of gasoline vehicular emissions on ambient VOCs should be pointed out.

Response: this sentence was reorganized, and this sentence was rewritten as (line 160-163 in new version) " The ratios of T/B, trans-/cis-2-butene, propane/n-butane and n-pentane/iso-pentane indicated that gasoline emission was dominated sources of ambient VOCs, and the source apportionment by PMF model result, and the detail description of source apportionment will be carried out in the following section."

4. Line 150: It should be "aerodynamic diameter"

Response: the description of diameter was corrected as"aerodynamic diameter". Line 176 in new version.

5. Line 228: Is the difference of photochemical reactions rate with OH radical between trans-2-butene and cis-2-butene large enough to compare?

Response: Trans-2-butene has higher photochemical reactions rate with OH radical in the atmosphere (kOH 6.40×10-11 s-1) than cis-2-buteneïijĹkOH 5.64×10-11 s-1ïijĽ, and the photochemical reactions rate with with OH radical of trans-2-butene is about 14% higher than that of cis-2-butene, and this difference would show in the photochemical reactions of these two congeners.

6. Line 278-281: The first two sentences are not necessary in conclusion part, moreover, key conclusions should be presented by supporting data.

Response: This comment was accepted, and the first two sentences were delated in the new version. And the supporting data was replenished in the new version in part of conclusion. And the part of conclusion was reorganized as "Comprehensive field work was carried out to investigate the origin and transform of VOCs within the dust-fine particles pollution periods in winter with the city of Xi'an. And the assumption of promotions of dust on the heterogeneous reactions of VOCs was further verified. Local vehicle exhaust (40%) and heating activities (41%) were found to be the most important sources of the ambient VOCs in Xi'an within winter, while long range transport air mass has limited impacts. Within the period of dust transport, loading of ambient VOCs decreased sharply from the late half period (average of 38 ppbv in dust period to average of 19 ppbv in transitional period), and the lowest concentration was observed in the transitional period (8 ppbv), in accordance with aging of primary VOCs. In addition, loading and proportion of secondary VOCs in gaseous phase and secondary ions and organic carbon in particulate phase increased with the aging of primary VOCs. Source strength, physical dispersion, and regional transport were eliminated from the major factor for the variation of the ambient VOCs. On another aspect, sharp increase of active metals concentrations (Ti and Fe) and fast decrease of trans-/cis-2-butene ratio was observed from the late half of dust transport period (1.21 to 0.65). In consequence, we conclude that windblown dust might accelerate the gas-solid heterogeneous reactions of atmospheric VOCs, and further induced the formation of SOA precursors." As shown in line 332-346 in new version. And the revised contents are marked in blue color in new version of manuscript.

---

## Author Response (AR1)

**Author's Response**

**1. Comments from Referees 1**

This manuscript presents a case study on the origin and transformation of ambient volatile organic compounds (VOCs) during an episode of dust-to-haze in a city in northwestern China. It presents the variations of VOCs, oxygenated VOCs, PM2.5 and its chemical components (i.e., ions, carbonaceous fractions and elements) during this interesting transition of dust-to-haze. It also highlights the possible reactions of VOCs under the influence of source contribution, ambient conditions as well as the heterogeneous reactions promoted by particle elements. The manuscript is well prepared and fits the scope of the journal. It could be considered for publication after the following concerns are clarified.

(1) The authors suggested the formation of secondary organic aerosols by the reactions of VOCs during the haze period. It could be true due to the significant increases of organic carbon during the haze period. However, I would suggest more efforts should be made to clarify this point. Somehow I would also suggest that it might be necessary to be highlighted in the text as well as in the title.

**Author's Response:** The suggestion is highly accepted, the point of formation of SOA by the VOCs during the haze period if replenished in the part of 3.4.

**Author's changes in manuscript:** Line 321-328 (in the new version), "A similar trend was seen for OC (Figure 6b), and content of particulate OC increased from 11.1 since dust event period to 47.1 in the haze period. In another aspect, the ratio of OC/EC increased from 1.3 to 4.9 in the dust-to-haze episode. The previous studies on the characterization of particles from traffic emission reported OC/EC values in the range of 0.28 to 0.92 in the diesel vehicles, and the OC/EC values were reported >2 in the gasoline vehicles. In addition, the OC/EC was reported in the range of 0.9 to 1.6 in the urban region in the city of Guangzhou. In the present study, the consistent increase of OC/EC would prove the formation of SOA in the dust-to-haze episode".

(2) In the part of Section 3.2, I found it was quite difficult to read through this part. The influences of sources (seems not explicit), boundary layer, long-range transport and photochemical reactions were discussed but not in a clear manner, which made me very difficult to follow the conclusions.

**Author's Response:** the suggestion is accepted, and this part is reorganized, as seen in section 3.2.

**Author's changes in manuscript:** As seen in line 234-235, "To evaluate the impact of sources types on the variation of VOCs in the dust-to-haze episode, diurnal variation of VOCs was depicted" was replenished. Line 243-247, "In addition, winter heating activities was relatively active because of low temperatures during the transitional period, and this limited the possibility of reduced emission amounts. Hence the variation of source strength was eliminated from the major factor caused the extremely low concentration and relative aged composition of ambient VOCs" was replenished. Line 247, "Variation of physical dispersion was also eliminated." was replenished.

(3) L224-226: The variations of the ratios of trans/cis-2-butene are discussed here.

The finding/suggestion by the authors seems not really true if we have a close look at Fig. 4. For example, the increase in the rush hour seems not always the case during this observation. The increases could also be observed during the time periods of 19:00-22:00 and 11:00. In this case, it should be more careful to draw any conclusions e.g. to highlight the importance of photochemical reactions and promotion of dust particles.

**Author's Response:** In the present study, fast decrease of the ratios of trans/cis-2-butene in the dust-to-haze episode. Correlation analysis of trans/cis-2-butene verse sampling time was done in the dust-to-haze episode, and significant correlation were observed. As mentioned in the comment, scatter peak of the ratios of trans/cis-2-butene were observed in the periods of 19:00-22:00 and 11:00. In our previous work of the characterization of VOCs in the roadside environment, peaks of ambient VOCs were sometimes observed in 8h-12h, which indicated tardive rush hours in the city of Xi'an. In addition, VOCs peaks were always observed in the midnight, and relative high density of high duty trucks that were used for Construction waste collection. Hence, the traffic emission of VOCs in the city of Xi'an is relative specific, which would be the reason of scatter peaks of the ratios of trans/cis-2-butene were observed in the periods of 19:00-22:00 and 11:00.

**Author's changes in manuscript:** In the new version of manuscript, correlation analysis of trans/cis-2-butene verse sampling time was done in the dust-to-haze episode, and significant correlation were observed with a $R^2$ of 0.6, and with a slope of 0.027/h, in spite of the interference of rush hour emission (Figure 4).

(4) As mentioned above, the linkage of VOCs to PM2.5 and its chemical components in Section 3.4 should be discussed in a more clear and explicit manner. I think catalysis of particle metal could be very important during the formation of haze and it should be studied here. By the way, I found Cl- was elevated at night (see Fig. 6) and it might show the contribution of biomass burning in northern areas of China.

**Author's Response:** we also believe the catalysis of particle metal could be very important during the formation of haze, and the linkage if VOCs to PM2.5 was further replenished in the section 3.4 (line 321-328). As mentioned by the reviewer, the content of Cl- was indeed higher in the night than daytime, this should be ascribed to heating activities (biomass or coal combustion), and this description was replenished in line 311.

**Author's changes in manuscript:** The linkage if VOCs to PM2.5 was further replenished in the section 3.4 (line 321-328) as mentioned in the second comment.

(5) L102-106: Quartz filter seems not an ideal filter media for XRF analysis due to the uneven surface. The authors should present more details of their XRF analysis.

**Author's Response:** In the present study, Quartz filter were used for $PM_{2.5}$ samples collection, due to the demand of analysis of carbon content and water-soluble ions. While quartz filter could not be used for analyzing of the content of some element with XRF, like Si. As a result, the content of particulate Si in the present study was not reported.

(6) minor mistakes: L158: It should be "the potential sources. . .are characterized". L241: "through" should be deleted.

**Author's Response:** the comment was fully accepted, and further improvement was done in line 184-185 in new version of manuscript.

**2. Comments from Referees 2**

The manuscript entitled "Origin and Transformation of Ambient VOCs during a Dust-to-Haze Episode in Northwest China" discussed the characteristics of ambient VOCs in a northwestern city in China, and the transformation of VOCs during a dust-to-haze episode was explored in this study. Generally, the paper is well organized and presented, and shows the possibility of VOC transformation through heterogeneous reactions during the episode. The paper can be considered for publication after the following minor revisions are made. Some specific comments are listed below.

1. Line 73: Change "sampling" to "samples"

**Author's Response:** This sentence was corrected.

**Author's changes in manuscript:** Line 81 in new version, "Severe dust-to-haze episode was observed in Xi'an and the surrounding areas from 8 November to 12 November in 2016, and samples was continuously collected during this period to investigate the chemical compositions of both VOCs and fine PM."

2. Line 82: Change "with" to "by"

**Author's Response:** This sentence was corrected.

**Author's changes in manuscript:** Line 93 in new version, this sentence was changed to "$PM_{2.5}$ filter samples were sampled with mini-volume samplers (Model Mini-Vol, Air Metrics Co., Oregon, USA) by a flow rate of 5 L $min^{-1}$".

3. Line 139 The last sentence is not clear. The contribution of gasoline vehicular emissions on ambient VOCs should be pointed out.

**Author's Response:** This sentence was reorganized.

**Author's changes in manuscript:** Line 160-163 in new version, this sentence was rewritten as "The ratios of T/B, trans-/cis-2-butene, propane/n-butane and n-pentane/iso-pentane indicated that gasoline emission was dominated sources of ambient VOCs, and the source apportionment by PMF model result, and the detail description of source apportionment will be carried out in the following section."

4. Line 150: It should be "aerodynamic diameter"

**Author's Response:** This sentence was corrected.

**Author's changes in manuscript:** Line 176 in new version, "the description of diameter was corrected as"aerodynamic diameter"".

5. Line 228: Is the difference of photochemical reactions rate with OH radical between trans-2-butene and cis-2-butene large enough to compare?

**Author's Response:** Trans-2-butene has higher photochemical reactions rate with OH radical in the atmosphere ($k_{OH}$ $6.40 \times 10\text{-}11$ $s^{-1}$) than cis-2-butene ($k_{OH}$ $5.64 \times 10\text{-}11$ $s^{-1}$) , and the photochemical reactions rate with with OH radical of trans-2-butene is about 14% higher than that of cis-2-butene, and this difference would show in the photochemical reactions of these two congeners.

6. Line 278-281: The first two sentences are not necessary in conclusion part, moreover, key conclusions should be presented by supporting data.

**Author's Response:** This comment was accepted, and the first two sentences were delated in the new version. And the supporting data was replenished in the new version in part of conclusion. And the part of conclusion was reorganized.

**Author's changes in manuscript:** Line 332-346 in new version, "Comprehensive field work was carried out to investigate the origin and transform of VOCs within the dust-fine particles pollution periods in winter with the city of Xi'an. And the assumption of promotions of dust on the heterogeneous reactions of VOCs was further verified. Local vehicle exhaust (40%) and heating activities (41%) were found to be the most important sources of the ambient VOCs in Xi'an within winter, while long range transport air mass has limited impacts. Within the period of dust transport, loading of ambient VOCs decreased sharply from the late half period (average of 38 ppbv in dust period to average of 19 ppbv in transitional period), and the lowest concentration was observed in the transitional period (8 ppbv), in accordance with aging of primary VOCs. In addition, loading and proportion of secondary VOCs in gaseous phase and secondary ions and organic carbon in particulate phase increased with the aging of primary VOCs. Source strength, physical dispersion, and regional transport were eliminated from the major factor for the variation of the ambient VOCs. On another aspect, sharp increase of active metals concentrations (Ti and Fe) and fast decrease of trans-/cis-2-butene ratio was observed from the late half of dust transport period (1.21 to 0.65). In consequence, we conclude that windblown dust might accelerate the gas-solid heterogeneous reactions of atmospheric VOCs, and further induced the formation of SOA precursors."

**3.  Short comment  1**

The current manuscript organized well. But some places may be improved in future:

1. Why did the authors choose the period of dust-to-haze episode to observe? How did it relate with the reaction of aerosols? Please describe the relationship between the scientific issue and the research design.

**Author's Response:** Dust is one of the most important sources of airborne metallic oxides in the natural environment, and the previous studies found explosive growth of ultrafine particles following dust episode, which implied positive effect of dust on the formation of fine particles. In addition, controlled experiments in chamber found surface transitional- metal-catalyzed chain reaction would highly impact formation of organic and inorganic aerosols on the surface of mineral dust (this was introduced in the part of introduction). Hence, dust was viewed to highly impact the transformation of organic and inorganic gas in the atmosphere. While few field work on the transformation of ambient VOCs in the dust event were found in the previous studies to support the assumption that dust would impact the transformation of VOCs in the ambient VOCs. The present study was designed to investigate the transformation of VOCs in dust event, and this work might helpful on further understanding of the impacts of dust on the transformation of VOCs and its further impact on the formation of fine particles.

2. In the Results and Discussions, the basic results, such as the levels of pollutants and the ratios of compounds, were not shown the manuscript. It is strange for readers that the sections starts with the sentences like "the concentrations in Xi'an are similar with Beijing" or "the T/B ratios in this study indicated that. . ."

**Author's Response:** This comment is fully accepted. In our manuscript, Table S2 were given in the supporting information, and the level and composition of ambient VOCs was summarized in this table. We had further improved our manuscript in this part, and the detail description of VOCs level and composition was replenished in the next version of manuscript.

**Author's changes in manuscript:** Line 135-140, "In the present study, mixing ratios of the sum of non-methane hydrocarbon was $36.0 \pm 15.7$ppbv, which was lower comparing to that in Beijing and Guangzhou with values of 51.0 and 47.8 ppbv, respectively. Similar levels of alkenes were seen at the cities of Beijing (9.4 ppbv) and Guangzhou (8.2 ppbv) comparing to that in the present study (9.2 ppbv, Table S2, Ho et al., 2004; Liu et al., 2008b). Unexpectedly, the aromatics were slightly higher in Xi'an (10.3 ppbv) than that in Beijing (9.6 ppbv), and 50% higher than that in Guangzhou (6.8 ppbv, Shao et al., 2009; Zou et al., 2015)."

3. I don't think it is reasonable to use HYSPLIT model in L157-186. It had discussed that the pollution come from local emission in the above sections.

**Author's Response:** Thanks very much for this comment. In the present study, origin of the ambient VOCs was analyzed in two aspects. In the first aspect, emission source types of VOCs were estimated with PMF model. In another aspect, effect of transport or local emission on the ambient VOCs were evaluated based on the variation of VOCs ratios and air mass transport. In this part, back trajectory of air mass in different period of sampling were used. Hence, the author believe HYSPLIT model should be useful in this part.

4. The abstract and Introduction highlighted the effect of metallic oxides on the photo-chemical reactions of VOCs. But in the manuscript, this situation wasn't discussed in details, and even no relationship figures of VOCs and metals were shown.

**Author's Response:** The objective of this work is to supply a reference of evidence the effect of dust on the transformation ambient VOCs and their further impact on the formation of secondary aerosol. While we believe in this study, our field work could just supply a solid transformation of ambient VOCs in the dust episode, which should be just a lateral evidence of particulate metallic oxides' catalysis effect on the oxidation of VOCs, be- cause too many factors impact the variation of level and composition of ambient VOCs. In addition, because the sampling time of VOCs and particles were not synchronous. The VOCs were sampled hourly, while the particles were only collected with two samples in 24 hours. As a result, we can't get a correlation analysis result of particulate metals verse VOCs. And in the following study, we will carry out fully chamber work on the transformation of VOCs on the dust surface that might supply direct evidence of dust catalysis effect on the transformation of VOCs.

[revised manuscript text omitted]

---

## Referee Report (RR1)

---- simple calculation -----

There is no explanation about concentration of trans-2-butene and cis-2-butene during the dust period, but just roughly estimate 0.72ppb and 0.6ppb at fresh pollution period (from 1-3-butadiene in Figure S5, Figure S1, and trans-2-butene/cis-2-butene ratio. 1-3-butadiene will similar source (car exhaust) and removal reaction by OH (k = 6.7E-11 cm$^3$/molecule/s) as trans and cis-2-butenes). 1-3-butadiene was 0.2 ppb at 20:00-24:00 on Nov.9 and decrease 0.05ppb at 20:00-24:00 on Nov.10 (decrease 1/4).

In the simple calculation, when original fresh polluted air (0.72ppb t-2-butene and 0.6ppb cis-2-butene) decrease to 0.15 ppb (about 1/4), the butene ratio ([trans-2-butene]/[cis-2-butene]) decrease from 1.2 to 1.0. It seems to be difficult to explain the observed butene ratio decrease (from 1.2 to 0.7).

---

## Author Response (AR2)

**Author's Response**

**1. Comments from Referees 1**

(1) L204-209: There are discussion xylene/ ethylbenzene. It would be better to show reaction rate constant of xylene and ethylbenzene with OH somewhere (1.9E-11 cm3/molecules/s, 0.7E-11 cm3/molecules/s).
**Author's Response:** The suggestion is highly accepted, the reaction rate constant of xylene and ethylbenzene with OH ($K_{OH}$) were replenished in the manuscript. In addition, the reaction rate constant of toluene and benzene with OH ($K_{OH}$) were replenished (line 197-199 in new version).

(2) L208: "X/B" -> "X/E"
**Author's Response:** corrected in the new version (line 206 in new version).

(3) L267,268: Unit of reaction rate constant is "cm3/molecule/s"
**Author's Response:** corrected in the new version (line 270 in new version, and line 197-199)

(4) L270-271: "morning rush hours 07:00-10:00) (Figure 4)". In Figure4, [trans-2-butene]/[cis-2-butene] is not high during 07:00-10:00 (on Nov.11). Relatively high t/c ratio was observed around 10:00-14:00 on Nov. 11.
**Author's Response:** the diurnal variation of [trans-2-butene]/[cis-2-butene] ratio in Nov. 11 was different from other sampling days, and the [trans-2-butene]/[cis-2-butene] ratio increased from 9:00, and reached peak at 12:00. And Nov. 11 was Saturday, and the morning rush hours might be postponed in the weekend. And in our previous study (Li et al. 2017, Atmospheric Environment, doi.org/10.1016/j.atmosenv.2017.04.029), peak of TVOCs were sometime found in the noon due to higher vehicle number and higher evaporation of fuel.

(5) L311: "Cl-1" -> "C-"
**Author's Response:** corrected in the new version.

(6) L325: Remove extra "()"
**Author's Response:** corrected in the new version.

(7) Fig.5 and Fig.6 Y-axis: "percent (%)" -> "proportion of concentration (%)" or "fraction (%)" etc.
**Author's Response:** corrected in figure 5 and figure 6 (page 24, page 25).

(8) Figure S1: It is better change X and Y axis of cis-2-butene vs trans-2-butene plot. (in the main text, the ratio indicated as [trans-2-butene]/[cis-2-butene].) The slope of [trans-2-butene]/[cis-2-butene] = 1.19 (1/0.84).
**Author's Response:** corrected in figure S1.

(9) Figure S5 x-axis: There are two Nov.11 (right "Nov. 11" -> "Nov.12").
**Author's Response:** corrected in figure S5.

(10) ---- simple calculation -----There is no explanation about concentration of trans-2-butene and cis-2-butene during the dust period, but just roughly estimate 0.72ppb and 0.6ppb at fresh pollution period (from 1-3-butadiene in Figure S5, Figure S1, and trans-2-butene/cis-2-butene ratio. 1-3-butadiene will similar source (car exhaust) and removal reaction by OH (k = 6.7E-11 cm3/molecule/s) as trans and cis-2-butenes). 1-3-butadiene was 0.2 ppb at 20:00-24:00 on Nov.9 and decrease 0.05ppb at 20:00-24:00 on Nov.10 (decrease 1/4). In the simple calculation, when original fresh polluted air (0.72ppb t-2-butene and 0.6ppb cis-2-butene) decrease to 0.15 ppb (about 1/4), the butene ratio ([trans-2-butene]/[cis-2-butene]) decrease from 1.2 to 1.0. It seems to be difficult to explain the observed butene ratio decrease (from 1.2 to 0.7).
**Author's Response:** it is a interesting calculation, and this maybe helpful for further understanding of the photochemical reactions of ambient VOCs. And the composition of ambient VOCs is also impacted by the variation of source strength, and it was found that [1-3-butadiene]/[butene] ratio varies significantly among the sources of biomass burning, gasoline exhaust and diesel exhaust (Liu et al. 2008, Atmospheric Environment, doi:10.1016/j.atmosenv.2008.01.070 ). And biomass burning and traffic emission were important sources of ambient VOCs in the sampling period, hence the variation of emission strength might cause the inconsistency between the calculation result and the observed result.

---

## Author Response (AR3)

**Author's Response**

Technical comments from editor:

1. Line 19. … roles of the oxidation of volatile organic compounds (VOC).
Response: corrected in line 19 in the new version, and the abbreviation of VOCs was added.

2. Lines 23, 24, 167, 168 and others. Use three significant numbers. The accuracy of the analytical methods used is not so high to use four significant numbers.
Response: corrected in the new version, the significant number were limited to 2 significant numbers for the sources apportionment result, as seen in line 23, line 24, line 168, and line 169 in the new version.

3. Lines 28, 44 and others. …of iron (Fe) and…
Response: corrected in line 28 and line 45, and the "Iron" was corrected as "iron".

4. Line 37, A phrase of "could cause the coating thickness of black carbon" is not clear. It needs a rephrase.
Response: the phrase was reorganized as "OH-initiated oxidation of m-xylene was found to cause the coating thickness of black carbon, which further induced the increase of particle size (1.5 to 10.4 times) and effective density (from 0.43 to 1.45 g $cm^{-3}$) ", as seen in line 38-40 in new version.

5. Line 56. Co-existent heterogenous.
Response: corrected, line 57 in the new version.

6. Line 88. …local time (LT) and..
Response: corrected, line 89 in the new version.

7. Line 122, 155, 213, 280, 282, 289, 315, 316 and others. Add a space in between references and other places. …5% (Ho et al., 2017; Ho et al., 2018).
Response: corrected, line 66, line 68, line 183,line 214, line 280, line 282, line 289, line 315 and line 316.

8. Line 132. …mixing ratio of the…
Response: corrected, line 133 in the new version.

9. Line 136. …, the levels of aromatics were…
Response: corrected, line 137 in the new version.

10. Line 160. …was a dominant source of…
Response: corrected, line 161 in the new version.

11. line 179. use a superscript for ug m-3.
Response: corrected, line 180 in the new version.

12. Line 196. Benzene
Response: corrected, line 197 in the new version.

13. Line 220. .., levels of ambient VOCs….
Response: corrected, line 221 in the new version.

14. Line 238. Remove "And".
Response: corrected, line 239 in the new version.

15. Line 241. ..rush hours…
Response: corrected, line 241 in the new version.

16. Line 250. …relatively high..
Response: corrected, line 250 in the new version.

17. Line 256. …relatively active VOCs..
Response: corrected, line 256 in the new version.

18. Line 257. ..would be aged with..
Response: corrected, line 257 in the new version.

19. Line 263. …of the VOC isomerides were…
Response: corrected, line 263 in the new version.

20. Line 294. X/E ratios were..
Response: corrected, line 294 in the new version.

21. Line 326. OC/EC ratios
Response: corrected, line 326-328.

22. Line 327. In addition, the OC/EC ratios were reported…
Response: corrected, line 326-328 in the new version.

23. Line 333. Summary and Conclusion
Response: corrected.

24. Figures 5 and 6. Use a capital for Proportion of …, and Fraction (%).lt.
Response: corrected.

[revised manuscript text omitted]